# Morphing Wing Based on Trigonal Bipyramidal Tensegrity Structure and Parallel Mechanism

**Jian Sun** [1,2], **Xiangkun Li** [1,2], **Yundou Xu** [1,2,*], **Tianyue Pu** [1,2], **Jiantao Yao** [1,2] and **Yongsheng Zhao** [1,2]

[1] Parallel Robot and Mechatronic System Laboratory of Hebei Province, Yanshan University, Qinhuangdao 066004, China
[2] Key Laboratory of Advanced Forging and Stamping Technology and Science of Ministry of National Education, Yanshan University, Qinhuangdao 066004, China
* Correspondence: ydxu@ysu.edu.cn; Tel.: +86-335-807-8704

**Abstract:** The development of morphing wings is in the pursuit of lighter weight, higher stiffness and strength, and better flexible morphing ability. A structure that can be used as both the bearing structure and the morphing mechanism is the optimal choice for the morphing wing. A morphing wing composed of a tensegrity structure and a non-overconstrained parallel mechanism was designed. The self-balancing trigonal bipyramidal tensegrity structure was designed based on the shape-finding method and force-equilibrium equation of nodes. The 4SPS-RS parallel mechanism that can complete wing morphing was designed based on the configuration synthesis method. The degree of freedom and inverse solution of the parallel mechanism was obtained based on the screw theory, and the Jacobian matrix of the parallel mechanism was established. The stiffness model of the tensegrity structure and the 4SPS-RS parallel mechanism was established. The relationship between the deformation of the 4SPS-RS parallel mechanism and sweep angle, torsion angle, spanwise bending, and span was obtained. Through the modular assembly and distributed drive, the morphing wing could perform smooth and continuous morphing locally and globally. In the static state, it has the advantages of high stiffness and large bearing capacity. In the process of morphing, it can complete morphing motion with four degrees of freedom in changing sweep, twist, spanwise bending, and span of the wing.

**Keywords:** morphing wing; tensegrity structure; parallel mechanism; distributed drive; stiffness analysis



## 1. Introduction

The traditional aircraft is designed for a specific flight condition, and its shape cannot change. Thus, it can only maintain high flight efficiency in a specific condition to meet specific task requirements [1]. With the development of the aviation industry, the aircraft designed for specific flight conditions finds it difficult to meet mission requirements. Future aircraft need to achieve high flight performance and fuel efficiency in different flight conditions, thus producing less noise and vibration [2]. Birds in nature change the shape of their wings during flight to adapt to different flight conditions. Inspired by this, scientists have attempted to change the traditional aircraft structure, to make the aircraft adapt to different flight conditions and meet different mission requirements. At present, the morphing wing has attracted much attention as one of the most effective methods.

The morphing wing can be classified into three types: plane transformation (span, chord, sweep); out-of-plane transformation (twist, dihedral, spanwise bending), and airfoil adjustment (camber and thickness) [3]. Different aerodynamic performances can be achieved by setting different wing parameters. For example: in high-speed flight, increasing the sweep of the wing can increase the critical Mach number of the aircraft and generate a smaller shock resistance. Aircraft can perform rapid penetration and supersonic flight missions; meanwhile, in low-speed flight, reducing the sweep of wings can improve the

flight efficiency and maneuverability of the aircraft. Aircraft can perform low-speed flight and short-range take-off and landing missions. Increasing the span of wings can improve flight efficiency, reduce induced drag, and increase flight range. Furthermore, aircraft can perform climbing and low-speed cruise missions.

The morphing wing mechanism can change the shape of the wing and increase the weight of the wing. To avoid increasing flight efficiency offset by increasing aircraft weight, a morphing wing mechanism that can be used as a bearing structure and morphing mechanism was designed. In the static state, the wing has great stiffness and bearing capacity, and in the morphing state, the wing can morph smoothly and continuously. The main research content of this paper was to design a morphing wing with multiple degrees of freedom and high stiffness and bearing capacity to meet the requirements of morphing movement and structural lightweight.

Amin Moosavia [4–6] proposed the main beam structure of a morphing wing composed of 8SPS parallel mechanisms in series. The wing can achieve smooth and continuous morphing of 6 degrees of freedom, and it has high stiffness and bearing capacity in the static state. David Cleaver [7–9] proposed a tensegrity structure with four driving controls and three degrees of freedom, and the structural stability was verified under different driving lengths. Meanwhile, a form-finding method was proposed to determine the stability of the tensegrity structure and stiffness characteristics.

Guang Yang [10] presented a wing morphing mechanism with a variable sweep angle based on a parallelogram mechanism composed of parallelogram basic units. Zhe Hui [11] designed an asymmetric variable sweep wing based on the pigeon wing structure. It can simulate the flight attitude of pigeons and improve the aerodynamic characteristics of aircraft through morphing skeleton control.

Benjamin Jenett [12] proposed a modular assembly morphing wing with variable torsion angles. The wing is composed of lightweight modular building blocks that are simple to manufacture. Nguyen K. Pham [13] developed a wing morphing mechanism with a cylindrical tensegrity structure that can realize continuous torsion of the wing. The optimal parameters of the mechanism under different torsion angles were obtained by establishing a finite element analysis model. Bing Luo [14] presented a wing morphing mechanism composed of gears with different speed ratios to realize wing torsional deformation. It is driven by a motor with a compact structure, small inertia, and low control difficulty. Haibo Zhang [15] proposed a morphing wing with a modular cellular structure of non-uniform density to realize wing torsional deformation.

D. Matthew Boston [16] developed a variable span morphing wing composed of cellular metamaterials exhibiting multiple stable shapes. It allows the structure to produce large elastic deformation without losing bearing capacity. Muhammed S. Parancheerivilakkathil [17] designed a morphing wing with a span extension of 25%. The pitching angle at the end of the wing can be adjusted to alleviate gust load. Inspired by kingfisher wings, Zhong Yun [18] proposed a folding wing with span morphing based on Sarrus linkages. The folding link was optimized based on multibody dynamics. Rafic M. Ajaj [19] developed a morphing wing driven by gears and racks, which can perform symmetric or asymmetric span deformation. It can achieve up to 50% span deformation. Based on the maximum stiffness and minimum flexibility of the wing, T.-M. Dao [20] proposed an optimization method for wing span morphing.

Jieyu Wang [21,22] proposed a 3-DOF morphing mechanism driven by two locking actuators. In the static state, the mechanism can withstand loads as statically determinate truss structures. Wing rib morphing can be realized by modular assembly along the wing span. David H. Myszka [23] analyzed the strength and stiffness of a tensegrity structure and a rigid structure and evaluated the characteristics of aircraft wings. Then, a wing rib morphing mechanism based on tensegrity structure was proposed.

However, the traditional morphing wing has a large weight and driving force. To solve this problem, this paper proposes a morphing wing structure that is composed of tensegrity structures and parallel mechanisms. First, a self-balancing trigonal bipyramidal

tensegrity structure that is composed of four rigid rods and six prestressed flexible cables was designed, based on the shape-finding method and force-equilibrium equation of nodes. As a special truss structure, the tensegrity structure is featured with high stiffness and bearing capacity, and it also has lighter weight because of the use of flexible cables. It can meet the bearing requirements of the wing. The optimal basic unit structure of the tensegrity structure and the combinatorial unit layout are obtained based on the principle of maximum stiffness and minimum weight. Then, a novel 4SPS-RS parallel mechanism [24,25] that can complete wing morphing was designed, based on the configuration synthesis method. The mechanism can realize morphing motion with four degrees of freedom in changing the sweep, twist, spanwise bending, and span of the wing. Meanwhile, it improves the stiffness and bearing capacity of the connection position in the tensegrity structure. Since the 4SPS-RS parallel mechanism is not over-constrained, it has a lower driving number and lighter weight, and it can ably meet the morphing requirements of the wing. In the static state, the morphing wing inherits the advantages of high stiffness and large bearing capacity of the tensegrity structure and the parallel mechanism. In the process of morphing, it can perform a smooth and continued morphing motion of the wing. Finally, the stiffness matrix of the tensegrity structure and the parallel mechanism is established, and the relationship between the deformation of the 4SPS-RS parallel mechanism and the change in the sweep, twist, spanwise bending, and span is obtained.

The rest of this paper is as follows: In Section 2, a trigonal bipyramidal tensegrity structure is designed, and the basic unit structure and the layout of the combined unit of the tensegrity structure are optimized based on the form-finding method and the nodal force balance equation. Meanwhile, a 4SPS-RS non-overconstrained parallel mechanism is designed based on the configuration synthesis method. In Section 3, the degree of freedom and inverse solution to the parallel mechanism is obtained, and the Jacobian matrix is established based on the screw theory. In Section 4, the stiffness model of the tensegrity structure and the parallel mechanism is established. The correctness of the theoretical stiffness model is verified by ANSYS simulation. In addition, the relationship between the deformation of the 4SPS parallel mechanism and the change in the sweep, twist, spanwise bending, and span is obtained.

## 2. Configuration Design of Morphing Wing

### 2.1. Design of Tensegrity Structure

The truss structure has the advantages of high stiffness and light weight, which can meet the bearing requirements of the wing girder. As a special truss structure, the tensegrity structure places some rigid rods in the structure with prestressed flexible cables. A self-balancing stable state with high stiffness can be achieved by adjusting the geometric relationship between the rigid rod and the flexible cable. Furthermore, the stiffness of the tensegrity structure is adjustable, and it is determined by the prestressing of the flexible cable. The greater the prestress, the higher the stiffness of the tension structure, and the stronger the bearing capacity. In this study, the tensegrity structure is used as the girder of the wing because it can meet the bearing demand of the girder of the wing and it can reduce the weight of the structure. As the simplest space truss structure, the tetrahedral structure has high stiffness and bearing capacity, so the tetrahedral structure is selected as the basic unit of the tensegrity support structure. Since the regular tetrahedron structure cannot constitute a self-balanced tensegrity structure, the basic unit of the trigonal bipyramidal tensegrity structure composed of two regular tetrahedron structures is obtained through shape-finding analysis. Its node and component names are shown in Figure 1.

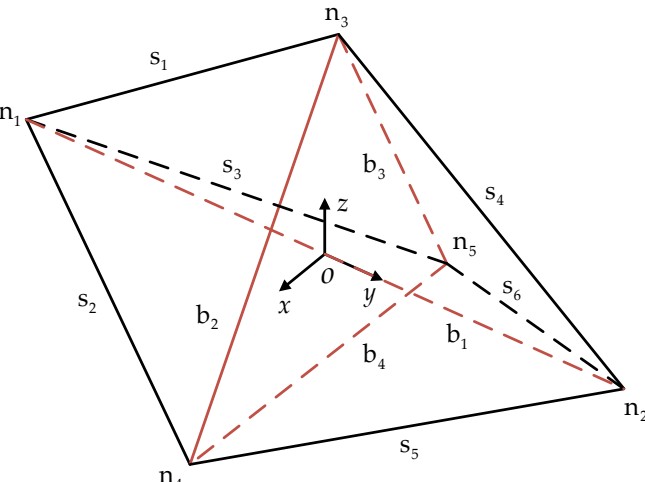

**Figure 1.** The trigonal bipyramidal tensegrity structure consists of six flexible cables and four rigid rods. The red line represents the rigid rod, and the black line represents the flexible cable with prestressing. The *o-xyz* coordinate system is established in the center position.

The position of each node can be represented by geometric relation as:

$$N = \begin{bmatrix} n_1 & n_2 & n_3 & n_4 & n_5 \end{bmatrix} \tag{1}$$

Then, the connection matrix of the rigid rod and flexible cable is established, and the position direction vector of each component is represented by the position coordinates of nodes:

$$C_B^{\mathrm{T}} = \begin{bmatrix} -1 & 0 & 0 & -1 \\ 1 & 0 & 0 & 0 \\ 0 & -1 & 0 & 0 \\ 0 & 1 & -1 & 0 \\ 0 & 0 & 1 & 1 \end{bmatrix} \tag{2}$$

$$C_S^{\mathrm{T}} = \begin{bmatrix} -1 & -1 & -1 & 0 & 0 & 0 \\ 0 & 0 & 0 & -1 & -1 & -1 \\ 1 & 0 & 0 & 1 & 0 & 0 \\ 0 & 1 & 0 & 0 & 1 & 0 \\ 0 & 0 & 1 & 0 & 0 & 1 \end{bmatrix} \tag{3}$$

$$B = NC_B^{\mathrm{T}} \tag{4}$$

$$S = NC_S^{\mathrm{T}} \tag{5}$$

In the tensegrity structure, each rigid rod only bears axial pressure, and the flexible cable only bears the axial tension. The force density is defined as the ratio of the internal force of the component to its initial length. The force density of rigid rod $b_i$ denoted as $\varphi_i$, and the force density of flexible cable $s_j$ is denoted as $\psi_j$. When the force of each node in the tensegrity structure is balanced, the structure is stable. So, the force equation of each node is established as:

$$A = \begin{bmatrix} N * C_S^{\mathrm{T}} * C_{S1}^* & -N * C_B^{\mathrm{T}} * C_{B1}^* \\ N * C_S^{\mathrm{T}} * C_{S2}^* & -N * C_B^{\mathrm{T}} * C_{B2}^* \\ \vdots & \vdots \\ N * C_S^{\mathrm{T}} * C_{S5}^* & -N * C_B^{\mathrm{T}} * C_{B5}^* \end{bmatrix} \tag{6}$$

$$F = A * \begin{bmatrix} \psi \\ \varphi \end{bmatrix} \tag{7}$$

where $C_{Si}^{*}$ denotes the diagonal matrix of the *i*-th column in matrix $C_S$, $C_{Bi}^{*}$ denotes diagonal matrix of the *i*-th column in matrix $C_B$.

If the node force equation has no solution, each node cannot be balanced. In this case, the node will produce displacement along the direction of the resultant force, and the structure is unstable. The force density of each component is calculated as follows:

$$\begin{cases} \psi_1 = \psi_2 = \psi_3 = \psi_4 = \psi_5 = \psi_6 \\ \varphi_2 = \varphi_3 = \varphi_4 \\ \varphi_1 : \varphi_{2,3,4} : \psi_{1,2,3,4,5,6} = 9 : 6 : 4 \end{cases} \tag{8}$$

The node force equation has only one solution. So, the trigonal bipyramidal tensegrity structure is stable. Based on the self-stress mode number and displacement mode number of the tensegrity structure, the stability of the structure is further proved:

$$\begin{cases} s = p + q - r_A \\ m = 3(n - k) - r_A \end{cases} \tag{9}$$

where *p* denotes the number of rigid rods, *q* denotes the number of flexible cables, *n* denotes the number of nodes, *k* denotes the number of constrained nodes, and $r_A$ denotes the rank of the equilibrium matrix.

When the number of constrained nodes is 2, the self-stress modal number and displacement modal number of the tensegrity structure are *s* = 1, *m* = 0. When *s* > 0, *m* = 0, the balance matrix *A* is a full-rank square matrix, and the tensegrity structure is statically indeterminate. Self-stress balance can be achieved by applying prestressing [26–28].

Taking the stiffness and weight of the tensegrity structure as the optimization objective, the layout of the combined units of the tensegrity structure was designed in this study. Without considering the effect of the 4SPS-RS parallel mechanism, the stiffness and weight of the morphing wing were analyzed by assuming that the basic unit of the tensegrity structure is rigid connections. Finally, the space center symmetry was the optimal layout.

### 2.2. Design of Wing Morphing Mechanism

From the morphing direction, the morphing wing can be classified into two types: morphing along the wing span direction (span, twist, dihedral, spanwise bending and sweep); morphing along the flight direction of the wing (chord, camber and thickness). Morphing wings with multiple morphing motions can meet different mission requirements. This paper focuses on the morphing along the wing spanwise direction. According to the characteristics of morphing motions, changing the span of the wing is a translational motion along the *x*-axis direction; changing the twist of the wing is a rotational motion around the *x*-axis; changing the dihedral angle and spanwise bending of the wing is a rotational motion around the *y*-axis; changing the sweep of the wing is a rotational motion around the *z*-axis. Therefore, morphing motions along the spanwise direction can be summarized as translational motion along *x*-axis and rotational motion around *x*, *y*, *z*-axis. As shown in Figure 2. Compared with the traditional driving method, the distributed driving structure can achieve smooth and continuous morphing in the whole and local wings. In summary, it was proposed to design a wing morphing mechanism that can achieve many wing morphing motions and meet many mission requirements [29–31].

In this study, a 4-DOF wing morphing mechanism that can change the sweep, twist, spanwise bending, and span of the wing was designed [32,33]. Due to the existence of flexible cables in the tensegrity structure, to ensure the rigid connection between the basic units of the tensegrity structure, the branch of the wing morphing mechanism should be arranged in the connection direction of the node. Meanwhile, since the basic units of the tensegrity structure were symmetrically arranged in the space center and there was no symmetrical relationship between nodes, the design scheme of the asymmetric parallel mechanism was adopted. As shown in Figure 3.

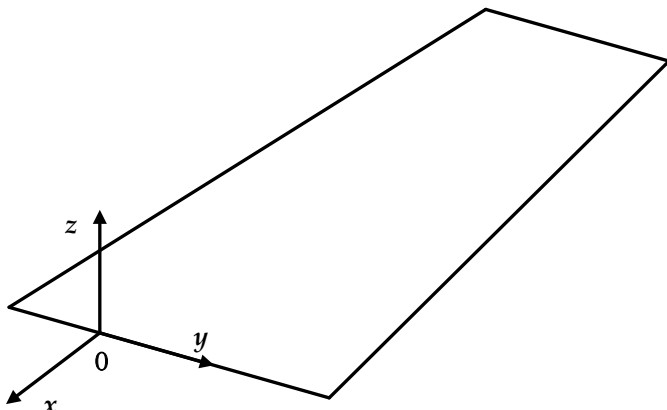

**Figure 2.** The morphing wing coordinate system diagram.

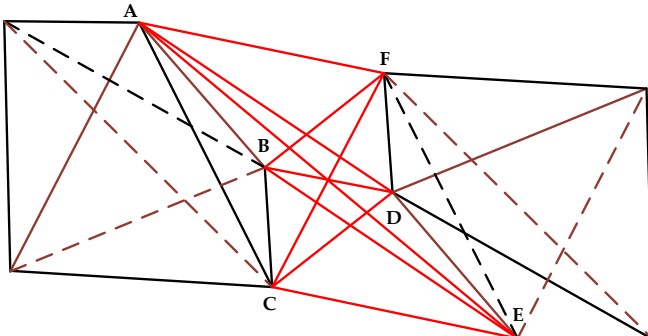

**Figure 3.** Possible branches of the wing morphing mechanism.

First, the position and direction of the central branch need to be determined. The three branches at the center of the wing deformation mechanism are AE, BD, and CF. According to the influence of the branch length, mechanism interference, rotation center position, and other factors, the CF branch was selected as the center branch, and the configuration design was conducted. The F position was selected as the rotation center, and the ball motion pair was added at this position. Therefore, the central branch provides three rotational degrees of freedom for the active tensegrity structure, and the driving branch was optimized according to the geometric position relationship of the nodes. AE and BD interfere with the central branch, while AF and BF provide driving forces through the rotation of the center, so they could not be used as driving branches. Finally, three of the four branches AD, BE, CD, and CE were selected as the driving branches.

The aircraft is mainly subjected to lift and drag from the air during flight. As shown in Table 1. So, three compression branches were selected as the driving branches according to the deformed direction of the four branches under load.

**Table 1.** The load direction of the driving branch.

| External Load | AD | BE | CD | CE |
|---|---|---|---|---|
| lift | compression | subject | subject | subject |
| resistance | compression | subject | compression | compression |

The BE branch is always in a tensile state. Therefore, AD, CD, and CE were set to SPS rigid drive branches. As shown in Figure 4.

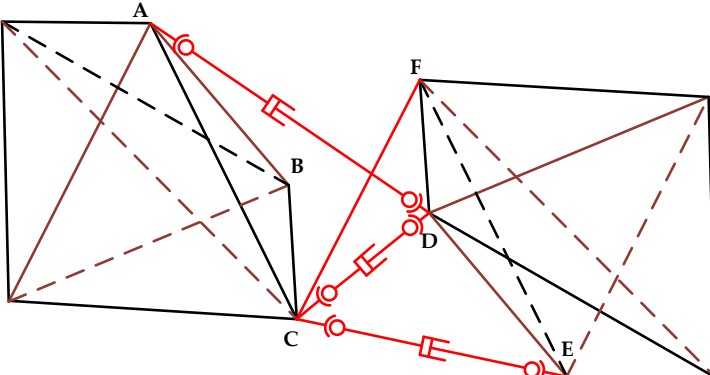

**Figure 4.** The center branch and drive branch axis of three rotational motions. The red circle represents the ball joint. The red square represents the mobile pair.

When node F performs translational motion, the wing achieves spanwise morphing. The central branch and the drive branch provide three binding forces that are not coplanar and intersect at a point for node F. The configurations of the central branch depend on whether the central branch is a drive branch. As shown in Figure 5.

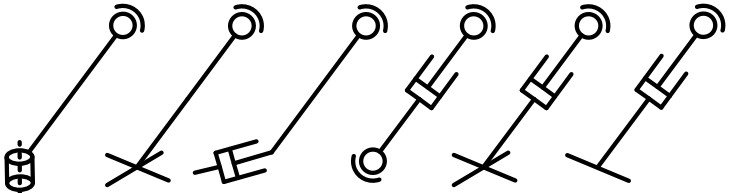

**Figure 5.** The central branch scheme. The black column and line represents the rotating pair. The black circle represents the ball joint. The black square represents the mobile pair.

Due to the weight limit of the wing, to reduce the quality of the morphing wing, the wing morphing mechanism should use fewer numbers of drives. Therefore, the central branch and the driving branch were chosen to provide a driving force and two passive binding forces. As a passive branch, the central branch provides two binding forces, and the driving branch provides a driving force. According to the configuration design, PS and RS met the design requirements of the central branch. Since the PS branch will make the bending moment at node C larger in the process of changing the span of the wing, RS was chosen as the central branch. Meanwhile, the BF branch was set to be the SPS rigid drive branch. As shown in Figure 6. The overall layout of the wing is shown in Figure 7.

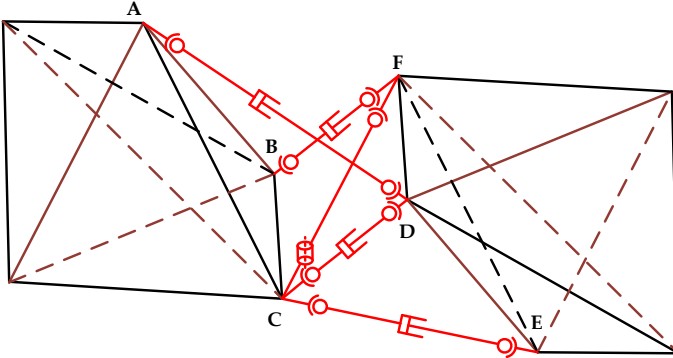

**Figure 6.** The sketch of the wing morphing mechanism. The red column represents the rotating pair. The red circle represents the ball joint. The red square represents the mobile pair.

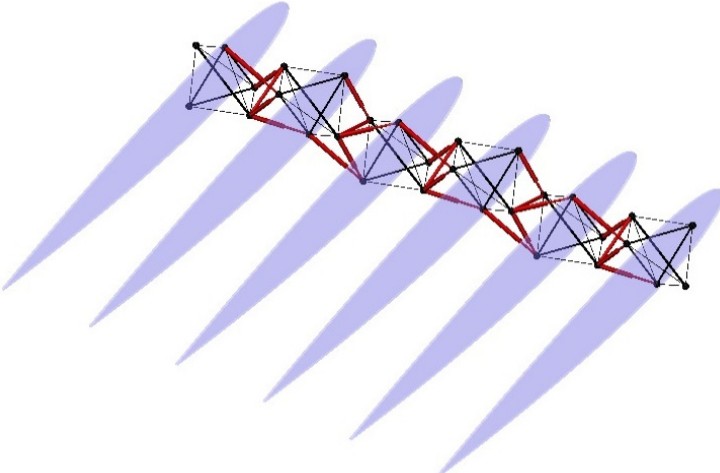

**Figure 7.** Three-dimensional figure of the morphing wing.

## 3. Kinematic Characteristics of the 4SPS-RS Parallel Mechanism

### 3.1. Degree of Freedom

The modified Kutzbach–Grübler formula was adopted to calculate the freedom of the wing morphing mechanism:

$$M = d(h - g) + \sum_{i=1}^{g} f_i + v \tag{10}$$

where $M$ denotes the mechanism freedom; $d$ denotes the rank of the mechanism; $h$ denotes the number of active components; $g$ denotes the number of kinematic pairs; $f_i$ denotes the degree of freedom of the $i$-th kinematic pair; $v$ is the number of overconstraints of the mechanism.

Since 4SPS-RS is a space parallel mechanism, the rank is 6. Meanwhile, two binding forces and four driving forces are linearly independent, so it is a non-overconstrained parallel mechanism. There are four local degrees of freedom in the driving branch, which should be reduced during the calculation. The number of degrees of freedom is calculated by data substitution:

$$M = 6(11 - 14 - 1) + 32 + 0 - 4 = 4 \tag{11}$$

### 3.2. Inverse Solution

A fixed coordinate system $O_G - x_G y_G z_G$ is established at the center of mass of the fixed tensegrity structure, a moving coordinate system $O_F - x_F y_F z_F$ is established at point F, and a moving coordinate system $O_H - x_H y_H z_H$ is established at the center of mass of the active tensegrity structure. The three coordinate systems have the same initial attitude. The $x$-axis is parallel to the flight direction of the aircraft, the $y$-axis is parallel to the span direction of the wing, and the $z$-axis is perpendicular to the wing plane and parallel to the rotational axis of the central branch. The attitude transformation matrix is established to convert the coordinates of points in the moving coordinate system to the fixed coordinate system. The mechanism sketch is shown in Figure 8.

$$\boldsymbol{P_G} = \boldsymbol{T}[\boldsymbol{P_H}, 1] \tag{12}$$

where $\boldsymbol{P_G}$ denotes the coordinates of point $\boldsymbol{P}$ in the fixed coordinate system, $\boldsymbol{T}$ denotes the attitude transformation matrix of the moving coordinate system relative to the fixed coordinate system, and $\boldsymbol{P_H}$ denotes the coordinates of point $\boldsymbol{P}$ in the moving coordinate system.

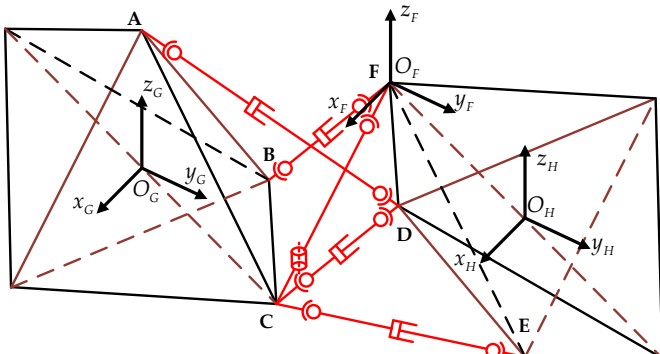

**Figure 8.** The sketch of the wing morphing mechanism and the establishment of coordinate systems. The red column represents the rotating pair. The red circle represents the ball joint. The red square represents the mobile pair.

The rotation transformation matrix adopts the Euler angle:

$$\boldsymbol{Rot} = \boldsymbol{Rot}(z, \gamma)\boldsymbol{Rot}(y, \beta)\boldsymbol{Rot}(x, \alpha) \tag{13}$$

The displacement transformation matrix is obtained as:

$$
\begin{cases}
\boldsymbol{Trans}(O_G, \mathrm{F}) = \begin{bmatrix} 1 & 0 & 0 & x_F \\ 0 & 1 & 0 & y_F \\ 0 & 0 & 1 & z_F \\ 0 & 0 & 0 & 1 \end{bmatrix} \\
\boldsymbol{Trans}(\mathrm{F}, O_H) = \begin{bmatrix} 1 & 0 & 0 & x_{OH} \\ 0 & 1 & 0 & y_{OH} \\ 0 & 0 & 1 & z_{OH} \\ 0 & 0 & 0 & 1 \end{bmatrix}
\end{cases} \tag{14}
$$

where $\boldsymbol{Trans}(O_G, \mathrm{F})$ denotes the displacement transformation matrix from $O_G$ to F, and $\boldsymbol{Trans}(\mathrm{F}, O_H)$ denotes the displacement transformation matrix from F to $O_H$.

Then, the $y$-axis coordinates of point F are determined. The coordinates of the $x$-axis and $z$-axis can be obtained according to the geometric relation:

$$x_F = -\left( \left| \sqrt{(L_{CF}\cos(\theta_{CF}))^2 - (y_F - y_C)^2} \right| - |x_C| \right) \tag{15}$$

$$z_F = L_{CF}\sin(\theta_{CF}) \tag{16}$$

where $L_{CF}$ denotes the distance between points $C$ and $F$ of the central branch; $\theta_{CF}$ denotes the angle between the straight line $CF$ and the $x_G - o_G - y_G$ plane; $x_C$ and $y_C$ denote the position coordinates of point $C$ in the $O_G - x_G y_G z_G$ coordinate system.

Then, the attitude transformation matrix of the 4SPS-RS parallel mechanism is established as:

$$\boldsymbol{T} = \boldsymbol{Trans}(O_G, \mathrm{F})\boldsymbol{Rot}(z, \gamma)\boldsymbol{Rot}(y, \beta)\boldsymbol{Rot}(x, \alpha)\boldsymbol{Trans}(\mathrm{F}, O_H) \tag{17}$$

The coordinates of nodes D, E, and F in the $O_H - x_H y_H z_H$ coordinate system are given. The coordinates in the coordinate system $O_G - x_G y_G z_G$ are calculated based on the attitude transformation matrix, and the inverse solution to the wing morphing mechanism can be obtained as:

$$
\begin{cases}
L_{CD} = |\boldsymbol{D_G} - \boldsymbol{C_G}| \\
L_{CE} = |\boldsymbol{E_G} - \boldsymbol{C_G}| \\
L_{BF} = |\boldsymbol{F_G} - \boldsymbol{B_G}| \\
L_{AD} = |\boldsymbol{D_G} - \boldsymbol{A_G}|
\end{cases} \tag{18}
$$

### 3.3. Jacobian Matrix

The joint motion spiral of the kinematic pair of each branch in the $O_H - x_H y_H z_H$ coordinate system is established as:

$$\$_{OH} = \sum_{i=1}^{5} \sum_{j=1}^{C_i} \theta_{i,j} \$_{i,j} \tag{19}$$

where $\theta_{i,j}$ denotes the amplitude of the $j$-th joint velocity of the $i$-th branch; $\$_{i,j}$ denotes the motion screw of the $j$-th joint of the $i$-th branch; $C_j$ denotes the number of joints of the $i$-th branch.

The attitude transformation matrix $T_H$ is transformed from the coordinate system $O_H - x_H y_H z_H$ to $O_G - x_G y_G z_G$ as:

$$T_H = \textbf{Trans}(O_H, \text{F})\textbf{Rot}(z, -\gamma)\textbf{Rot}(y, -\beta)\textbf{Rot}(x, -\alpha)\textbf{Trans}(\text{F}, O_G) \tag{20}$$

The coordinates of nodes A, B, and C in the $O_G - x_G y_G z_G$ coordinate system are given. The coordinates in the $O_H - x_H y_H z_H$ coordinate system are calculated based on the attitude transformation matrix $T_H$.

The degree of freedom of the CF branch is 4. Since the reciprocal product of the spiral is zero, the branch provides two binding forces:

$$\begin{cases} \$_{CF}^{f1} = \begin{bmatrix} 0 & 0 & 1 & ; & F_H \times \begin{pmatrix} 0 & 0 & 1 \end{pmatrix} \end{bmatrix} \\ \$_{CF}^{f2} = \begin{bmatrix} F_H - C_H & ; & F_H \times (F_H - C_H) \end{bmatrix} \end{cases} \tag{21}$$

The Jacobi matrix $J_y$ of constraint wrench is obtained as:

$$J_y = \begin{bmatrix} F_H \times \begin{pmatrix} 0 & 0 & 1 \end{pmatrix} & 0 & 0 & 1 \\ F_H \times (F_H - C_H) & F_H - C_H \end{bmatrix} \tag{22}$$

The reciprocal product of constraint wrench and joint twist is zero:

$$J_y \$_{OH} = 0 \tag{23}$$

The branches $CD$, $CE$, $BF$, and $AD$ have six degrees of freedom, so they do not provide constraint wrenches. Instead, they provide four driving forces as follows:

$$\begin{cases} \$_{CD}^{fq} = \begin{bmatrix} D_H - C_H & ; & D_H \times (D_H - C_H) \end{bmatrix} \\ \$_{CE}^{fq} = \begin{bmatrix} E_H - C_H & ; & E_H \times (E_H - C_H) \end{bmatrix} \\ \$_{BF}^{fq} = \begin{bmatrix} F_H - B_H & ; & F_H \times (F_H - B_H) \end{bmatrix} \\ \$_{AD}^{fq} = \begin{bmatrix} D_H - A_H & ; & D_H \times (D_H - A_H) \end{bmatrix} \end{cases} \tag{24}$$

The Jacobi matrix $J_q$ of driving wrenches is obtained as:

$$J_q = \begin{bmatrix} D_H \times (D_H - C_H) & D_H - C_H \\ E_H \times (E_H - C_H) & E_H - C_H \\ F_H \times (F_H - B_H) & F_H - B_H \\ D_H \times (D_H - A_H) & D_H - A_H \end{bmatrix} \tag{25}$$

The reciprocal product of the driving wrenches and joint twist is calculated as follows:

$$J_q \$_{OH} = \begin{bmatrix} \dot{d}_{CD} & \dot{d}_{CE} & \dot{d}_{BF} & \dot{d}_{AD} \end{bmatrix}^{\text{T}} \tag{26}$$

where $\dot{d}_{CD}$, $\dot{d}_{CE}$, $\dot{d}_{BF}$, and $\dot{d}_{AD}$ denote the velocity amplitude of the driving branch.

The Jacobian matrix of the 4SPS-RS parallel mechanism is composed of $J_y$ and $J_q$:

$$Q_V = JV \tag{27}$$

$$J = \begin{bmatrix} J_q & J_y \end{bmatrix}^{\mathrm{T}} \tag{28}$$

$$Q_V = \begin{bmatrix} \dot{d}_{CD} & \dot{d}_{CE} & \dot{d}_{BF} & \dot{d}_{AD} & 0 & 0 \end{bmatrix}^{\mathrm{T}} \tag{29}$$

$$V = \begin{bmatrix} \omega_x & \omega_y & \omega_z & v_x & v_y & v_z \end{bmatrix}^{\mathrm{T}} \tag{30}$$

The velocity Jacobian matrix of the 4SPS-RS parallel mechanism is established, and the mapping relationship between the driving input velocity and end position velocity is obtained [34,35].

According to the morphing mission requirements, the variation range of sweep is $[-\pi/9\sim0]$, the variation range of twist is $[-\pi/9\sim\pi/9]$, the variation range of spanwise bending is $[-\pi/9\sim\pi/9]$, and the variation range of span is $[0\sim50\text{ mm}]$. In order to make the 4SPS-RS parallel mechanism have better force performance, driving force stability and balance driving force maximum value are taken as optimization objectives.

Driving force stability describes the fluctuation of the driving force in the workspace.

$$\sigma = \sqrt{\frac{1}{n-1}\sum_{i=1}^{n}(f_i - f_a)} \tag{31}$$

where $f_i$ is the size of the driving force of branch i. $f_a$ is the average value of the driving force in the whole domain.

Balance driving forces of 4SPS-RS parallel mechanism are calculated, based on force Jacobian matrix.

$$Q_F = J^{-\mathrm{T}}F \tag{32}$$

$$Q_F = \begin{bmatrix} f_{CD} & f_{CE} & f_{BF} & f_{AD} & 0 & 0 \end{bmatrix}^{\mathrm{T}} \tag{33}$$

$$F = \begin{bmatrix} f_x & f_y & f_z & m_x & m_x & m_x \end{bmatrix}^{\mathrm{T}} \tag{34}$$

where $Q_F$ is the matrix of four driving forces. $F$ is the external load.

The three fixed nodes of the 4SPS-RS parallel mechanism form an equilateral triangle whose length is the side length of the tensegrity structure. Since the basic elements of tensegrity structures are identical, three active nodes form an identical equilateral triangle. The length of the equilateral triangle is $L_t$. The vertical height between the fixed surface and the active surface is $L_p$ In the workspace, two sizes are optimized. The variation range of sizes is given. The variation range of $L_t$ is [150~350]. The variation range of $L_p$ is [100~300]. These sizes are discretized. The moving platform moved 50 mm along $y$-axis, rotated 20 degrees around the $x$-axis and rotated 10 degrees around the $y$-axis and $z$-axis. The external loads are 100 N force along the $x$-axis and $y$-axis. The relationship between the maximum equilibrium driving force and the size is obtained. As shown in Figure 9.

Based on the above results, the mechanism size of the minimum balance driving force is $L_t$ = 150, $L_p$ = 100. Under the same $L_t$, changing the value of $L_p$, the mechanism will have a minimum equilibrium driving force. It can be obtained. As shown in Tables 2 and 3.

**Table 2.** Size combination 1–9.

|       | 1   | 2   | 3   | 4   | 5   | 6   | 7   | 8   | 9   |
|-------|-----|-----|-----|-----|-----|-----|-----|-----|-----|
| $L_t$ | 180 | 190 | 200 | 210 | 220 | 230 | 240 | 250 | 260 |
| $L_p$ | 100 | 110 | 120 | 120 | 130 | 130 | 140 | 140 | 150 |

**Table 3.** Size combination 10–18.

|        | 10  | 11  | 12  | 13  | 14  | 15  | 16  | 17  | 18  |
|--------|-----|-----|-----|-----|-----|-----|-----|-----|-----|
| $L_t$  | 270 | 280 | 290 | 300 | 310 | 320 | 330 | 340 | 350 |
| $L_p$  | 160 | 160 | 170 | 170 | 180 | 190 | 190 | 200 | 200 |

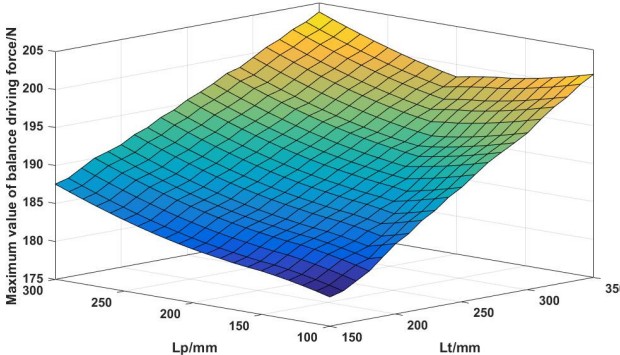

**Figure 9.** The relationship between the maximum value of balance driving force and the size.

Because the structure of the node position is complex, smaller dimensions cause interference in the structure. The size combination of $L_t$ = 300, $L_p$ = 170 was selected as the optimal size. Its driving force stability was 0.88. It meets mission requirements. The structure is shown in Figure 10.

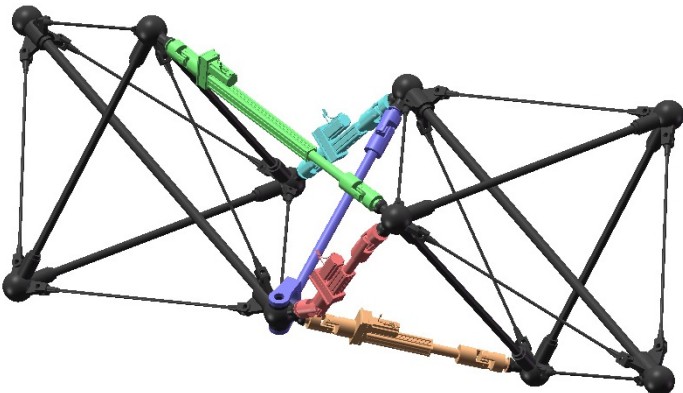

**Figure 10.** Three-dimensional figure of the basic unit of morphing wing.

## 4. Stiffness Analysis of Morphing Wing

The traditional wing has a large support stiffness to ensure that the wing will not produce large deformations during flight. When the main beam structure of the traditional wing is eliminated, the local stiffness of the wing may decrease. Thus, it is necessary to analyze and simulate the stiffness of the tensegrity structure and the parallel mechanism.

### 4.1. Stiffness Analysis of Tensegrity Structure

The node position of the tensegrity structure is shown in Figure 1. The angle between the projection of the connection line from node $n_1$ to node $n_2$ in the *x-o-y* plane and the positive direction of the *y*-axis is denoted as $\theta_{15}$. The unit vector of all components can be obtained from the node coordinates and connection matrix. The internal forces of each component are denoted as $f_{si}$ and $f_{bj}$. Then, the force at the nodes $n_3$ and $n_4$ caused by rod $b_2$ is obtained as:

$$\varphi_2 = f_{b2}/a \tag{35}$$

$$\begin{cases} f_{n3}^{b2} = -(\boldsymbol{n}_4 - \boldsymbol{n}_3)\varphi_2 \\ f_{n4}^{b2} = (\boldsymbol{n}_4 - \boldsymbol{n}_3)\varphi_2 \end{cases} \tag{36}$$

where a denotes the length of rigid rod $b_2$, and $\varphi_2$ denotes the force density value of rigid rod $b_2$.

The tangent stiffness matrix is constructed, and the node force component is derived from the node coordinate:

$$\begin{cases} \frac{\partial f_{n3i}^{b2}}{\partial n_{3j}} = -(n_{4i} - n_{3i})\frac{\partial \varphi_2}{\partial n_{3j}} + \delta_{ij}\varphi_2 \\ \frac{\partial f_{n4i}^{b2}}{\partial n_{3j}} = (n_{4i} - n_{3i})\frac{\partial \varphi_2}{\partial n_{3j}} - \delta_{ij}\varphi_2 \\ \frac{\partial f_{n3i}^{b2}}{\partial n_{4j}} = -(n_{4i} - n_{3i})\frac{\partial \varphi_2}{\partial n_{4j}} - \delta_{ij}\varphi_2 \\ \frac{\partial f_{n4i}^{b2}}{\partial n_{4j}} = (n_{4i} - n_{3i})\frac{\partial \varphi_2}{\partial n_{4j}} + \delta_{ij}\varphi_2 \end{cases} \tag{37}$$

where $f_{n3i}^{b2}$ represents the component of the force at position $n_3$ of the node caused by rigid rod $b_2$ in direction $i$; $f_{n4i}^{b2}$ represents the component of the force at position $n_4$ of the node caused by rigid rod $b_2$ in direction $i$; $n_{3j}$ represents the coordinate component of node $n_3$ in direction $j$; $n_{4j}$ represents the coordinate component of node $n_4$ in direction $j$.

When $i = j$, $\delta_{ij} = 1$; $i \neq j$, $\delta_{ij} = 0$. Through simplification, the following formula can be obtained:

$$\begin{cases} \frac{\partial \varphi_2}{\partial n_{3j}} = -\frac{1}{a}\left(\frac{\mathrm{d}f_{b2}}{\mathrm{d}a} - \varphi_2\right)b_{2j} \\ \frac{\partial \varphi_2}{\partial n_{4j}} = \frac{1}{a}\left(\frac{\mathrm{d}f_{b2}}{\mathrm{d}a} - \varphi_2\right)b_{2j} \end{cases} \tag{38}$$

According to the definition of tensile strength, we have:

$$\frac{\mathrm{d}f_{b2}}{\mathrm{d}a} = \frac{A_{b2}E_{b2}}{a} \tag{39}$$

where $A_{b2}$ represents the cross-sectional area of rigid rod $b_2$, and $E_{b2}$ denotes Young's Modulus of rigid rod $b_2$. Then, the following formula can be obtained:

$$H = \frac{\mathrm{d}f_{b2}}{\mathrm{d}a} - \varphi_2 \tag{40}$$

$$\begin{cases} \frac{\partial f_{n3}^{b2}}{\partial n_3} = \boldsymbol{b}_2 H \boldsymbol{b}_2^{\mathrm{T}} + \varphi_2 \mathbf{I} \\ \frac{\partial f_{n4}^{b2}}{\partial n_3} = -\boldsymbol{b}_2 H \boldsymbol{b}_2^{\mathrm{T}} - \varphi_2 \mathbf{I} \\ \frac{\partial f_{n3}^{b2}}{\partial n_4} = -\boldsymbol{b}_2 H \boldsymbol{b}_2^{\mathrm{T}} - \varphi_2 \mathbf{I} \\ \frac{\partial f_{n3}^{b2}}{\partial n_4} = \boldsymbol{b}_2 H \boldsymbol{b}_2^{\mathrm{T}} + \varphi_2 \mathbf{I} \end{cases} \tag{41}$$

For rigid rod $b_2$, the relationship between the small change in the displacement at the node position and the small change in the force at the node position can be obtained based on the tangent stiffness matrix:

$$\begin{bmatrix} \delta f_{n3}^{b2} \\ \delta f_{n4}^{b2} \end{bmatrix} = K_S^{b2} \begin{bmatrix} \delta \boldsymbol{n}_3 \\ \delta \boldsymbol{n}_4 \end{bmatrix} \tag{42}$$

$$K_S^{b2} = \begin{bmatrix} \boldsymbol{b}_2 \\ -\boldsymbol{b}_2 \end{bmatrix} [H] \begin{bmatrix} \boldsymbol{b}_2^{\mathrm{T}} & -\boldsymbol{b}_2^{\mathrm{T}} \end{bmatrix} + \begin{bmatrix} \varphi_2 \mathbf{I} & -\varphi_2 \mathbf{I} \\ -\varphi_2 \mathbf{I} & \varphi_2 \mathbf{I} \end{bmatrix} \tag{43}$$

The tangential stiffness matrix of rigid rod $b_2$ is obtained. Then, the tangential stiffness matrix of all components can be obtained through this deduction. The tangent stiffness matrix $K_S$ of the trigonal bipyramidal tensegrity structure is obtained by combining the tangent stiffness matrix of all components.

The active tensegrity structure is connected with the wing morphing mechanism through nodes $n_1$, $n_2$ and $n_3$. Since it is assumed that the wing morphing mechanism will not deform at $n_1$, $n_2$ and $n_3$, the tangent stiffness matrix is set to 1 at the diagonal position of lines 1, 2, 3, 7, 8, 9, 10, 11, 12, and the tangent stiffness matrix at other positions is set to 0. Based on this, the stiffness matrix under the initial boundary condition is obtained.

The main materials of aircraft wing bearing structure are 7075 aviation aluminum alloy and structural steel. Therefore, the influence of the two materials on stiffness was studied. The specific parameters of the 7075 aviation aluminum alloy are presented in Table 4.

**Table 4.** The parameter of the 7075 aviation aluminum alloy.

| Parameter | Density(kg/m$^3$) | Elastic Modulus (MPa) | Poisson Ratio |
|:---:|:---:|:---:|:---:|
| Value | 2810 | 71,000 | 0.33 |

Since there is no parameter selection for flexible cables in ANSYS, the existing Cable280 element was adopted to simulate the flexible cable. The material is structural steel. The specific parameters of the Cable280 unit are presented in Table 5.

**Table 5.** The parameter of the Cable280 unit.

| Parameter | Density (kg/m$^3$) | Elastic Modulus (MPa) | Poisson Ratio |
|:---:|:---:|:---:|:---:|
| Value | 1800 | 28,000 | 0.34 |

The sectional area of the rigid rob is $A_{b1} = 78.5$ mm$^2$, $A_{b2} = A_{b3} = A_{b4} = 50.3$ mm$^2$. The sectional area of the flexible cable is $A_{s1} = A_{s2} = A_{s3} = A_{s4} = A_{s5} = A_{s6} = 7.1$ mm$^2$. The length of the rigid rod and flexible cable can be obtained by the geometric relation. The internal force density ratio of each component is $\varphi_1 : \varphi_2 : \psi_1 = 9 : 6 : 4$, so $f_{b1} = 1470$ N, $f_{b2} = 600$ N and $f_{b3} = 400$ N. The rigid rod is compressed, and the flexible cable is pulled. Given a force of 100 N along the $z$-axis at node $n_2$, the theoretical deformation at the node $n_2$ and node $n_5$ can be calculated.

When the material is structural steel, the theoretical deformation is:

$$D_{LT}^{ST} = \begin{bmatrix} d_{Ln2}^{ST} \\ d_{Ln5}^{ST} \end{bmatrix} = \begin{bmatrix} 0.0321 & -0.0258 & 0.2034 \\ 0.0285 & 0.0347 & 0.0007 \end{bmatrix} \tag{44}$$

When the material is 7075 aviation aluminum alloy, the theoretical deformation is:

$$D_{LT}^{AL} = \begin{bmatrix} d_{Ln2}^{AL} \\ d_{Ln5}^{AL} \end{bmatrix} = \begin{bmatrix} 0.0308 & -0.0486 & 0.2931 \\ 0.0326 & 0.0336 & 0.0091 \end{bmatrix} \tag{45}$$

The deformation of each node position is obtained by ANSYS simulation. As shown in Figures 11 and 12.

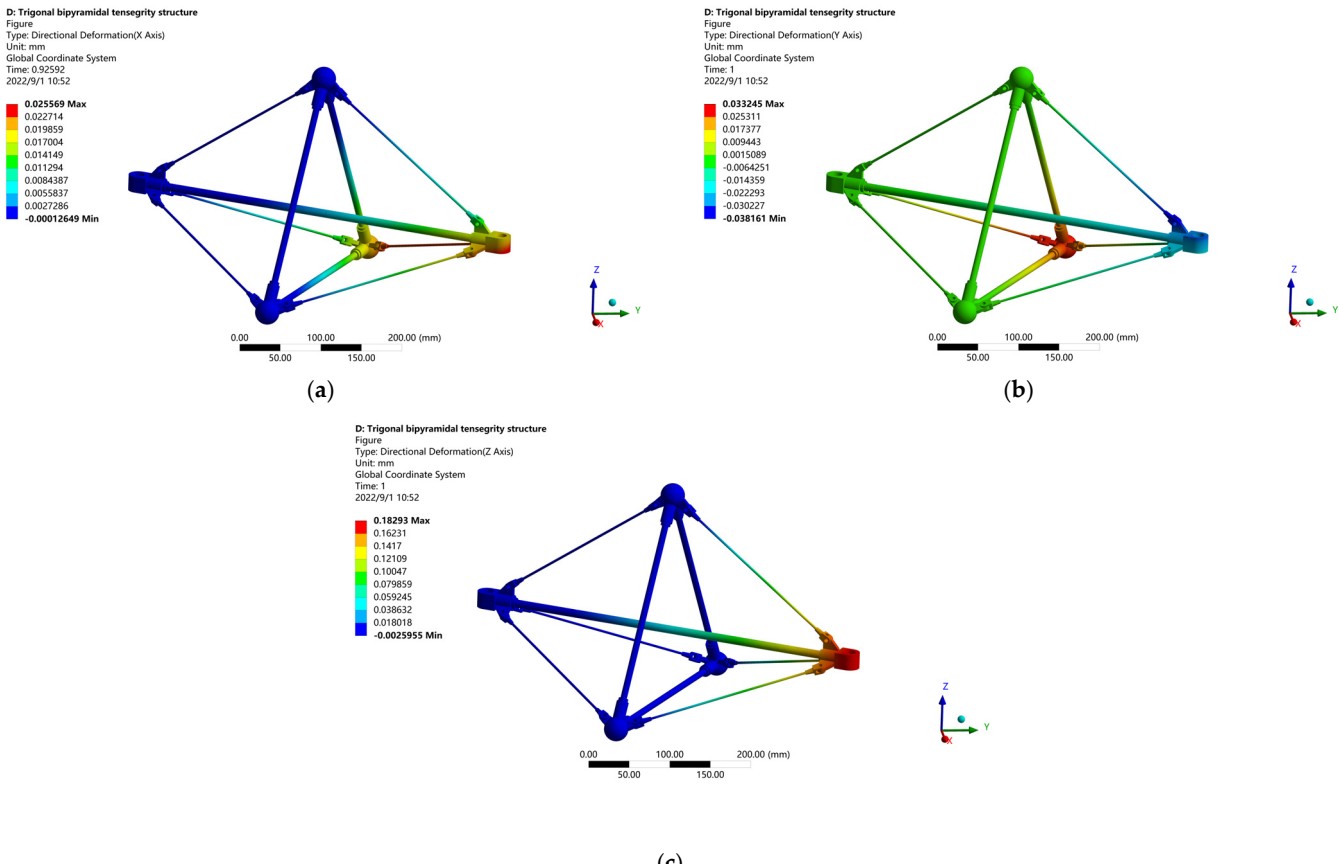

**Figure 11.** When the material is structural steel, simulation deformation of trigonal bipyramidal tensegrity structures in different coordinate directions: (**a**) *x*-axis; (**b**) *y*-axis; (**c**) *z*-axis.

The deformation along the coordinate axis at nodes *n*2 and *n*5 can be expressed. When the material is structural steel, the simulation deformation is:

$$D_{FT}^{ST} = \begin{bmatrix} d_{Fn2}^{ST} \\ d_{Fn5}^{ST} \end{bmatrix} = \begin{bmatrix} 0.0244 & -0.0247 & 0.1705 \\ 0.0229 & 0.0308 & 0.0009 \end{bmatrix} \tag{46}$$

When the material is 7075 aviation aluminum alloy, the simulation deformation is:

$$D_{FT}^{AL} = \begin{bmatrix} d_{Fn2}^{AL} \\ d_{Fn5}^{AL} \end{bmatrix} = \begin{bmatrix} 0.0225 & -0.0433 & 0.2467 \\ 0.0303 & 0.0321 & 0.0067 \end{bmatrix} \tag{47}$$

The above analyzes the influence of two materials on the structural stiffness, when the external load is the same, the deformation of 7075 aviation aluminum alloy is larger than that of structural steel. However, the weight of 7075 aviation aluminum alloy is lighter than that of structural steel. The combination of the two materials is the best option, with high-strength structural steel selected at key connection points and 7075 aviation aluminum alloy used in the wing body.

Due to the optimization design of some structures in the simulation and some meshing errors in the simulation, the deformation of the nodes obtained by the simulation was smaller than that obtained by the theoretical calculation. The average error was within 10%, which verifies the correctness of the stiffness theoretical model.

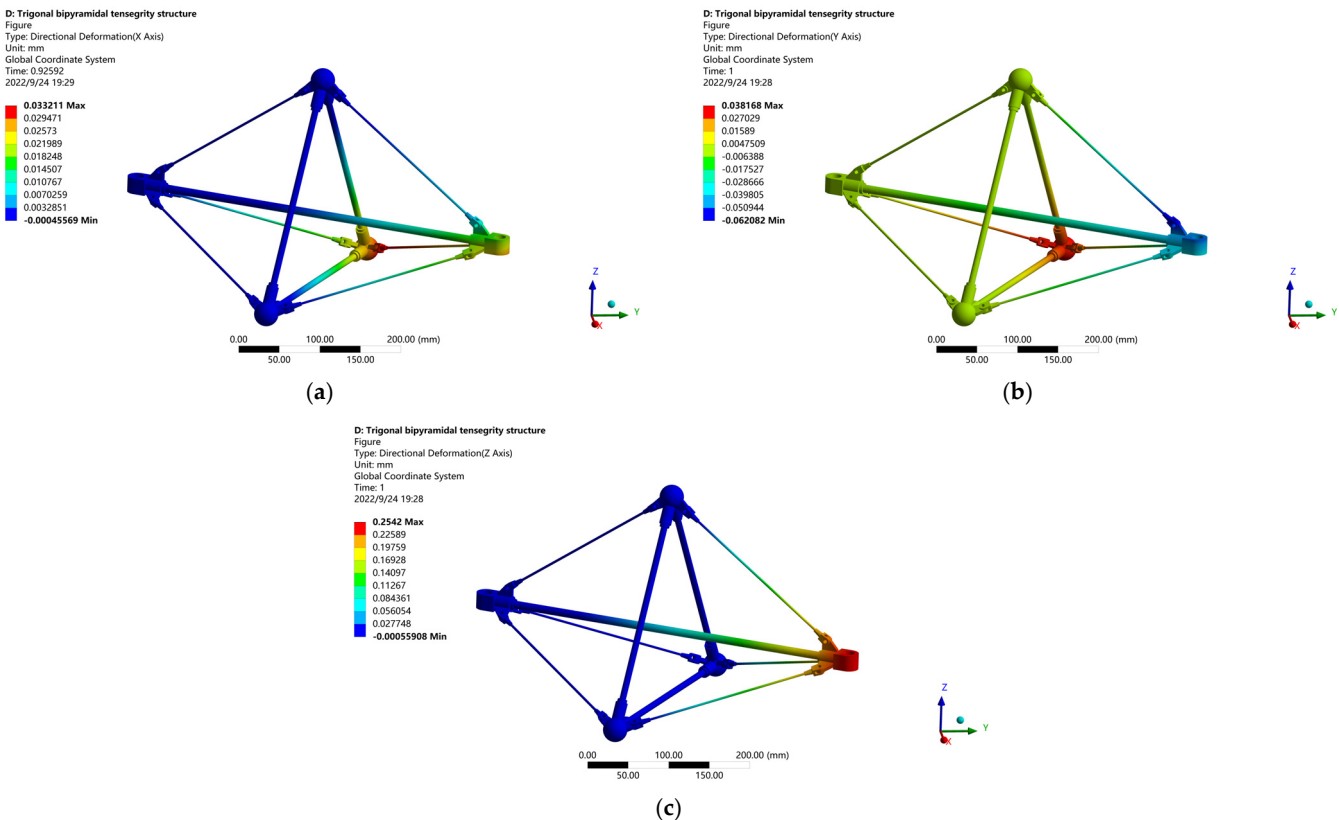

**Figure 12.** When the material is 7075 aviation aluminum alloy, simulation deformation of trigonal bipyramidal tensegrity structures in different coordinate directions: (**a**) *x*-axis; (**b**) *y*-axis; (**c**) *z*-axis.

### 4.2. Stiffness Analysis of the 4SPS-RS Parallel Mechanism

Here, the stiffness analysis model of the wing morphing mechanism is established. If the stiffness of the mechanism is poor, the deformation of the wing may be too large to generate the required lift. Since the analysis focuses on the morphing mechanism of the wing, it is assumed that the tensegrity structure is a rigid body. The influence of joint friction, environment, and other factors is ignored, and only the influence of the driving joint, passive joint and connecting rod on the stiffness of the mechanism is considered. Meanwhile, since SPS is used as the driving branch, the two bars in each branch are force bars, which only carry the load in the direction of the branch axis. Furthermore, in the driving branch, the support stiffness of the spherical pair is large, so its influence on the branch stiffness can be ignored. Thus, it is necessary to analyze the axial stiffness of the drive joint and the connecting rod. In the central branch, the bending and tensile stiffness of the central connecting rod are analyzed.

The drive branch stiffness model is established, and the ball screw is used as the drive, including the axial stiffness of the screw, the ball nut, and the connecting rod:

$$K_g = \frac{A_g E_g}{L_g} = \frac{\pi d_g^2 E_g}{4 L_g} \tag{48}$$

where $K_g$ represents the axial stiffness of the screw, $A_g$ represents the cross-sectional area of the screw, $d_g$ represents the thread diameter of the screw, $E_g$ represents the elastic model of the screw, and $L_g$ represents the distance between the load point and the double thrust bearing.

The stiffness analytical formula of the ball nut is obtained by referring to the national standard manual:

$$K_m = \frac{\pi i_m P_m E_m \tan^2 \theta_m \left( D_{m1}^2 - D_{m2}^2 \right)}{D_{m1}^2} \tag{49}$$

where $K_m$ represents the axial stiffness of the ball nut, $i_m$ represents the number of ball bearing circles, $P_m$ represents the ball screw guide, $E_m$ represents the elastic modulus of the ball nut, $\theta_m$ represents the contact angle, $D_{m1}$ represents the outer diameter of the ball nut, and $D_{m2}$ represents the diameter of the contact point on the ball nut.

The driving branch consists of two connecting rods. Connecting rod 1 connects the fixed tensegrity structure and the driving joint. It is used as the hollow rod because the internal needs to put the ball screw. Connecting rod 2 connects the active tensegrity structure and driving joint, and is used as the solid rod.

$$K_{L1} = \frac{E_1 \pi \left( D_1^2 - d_1^2 \right)}{4 L_1} \tag{50}$$

$$K_{L2} = \frac{E_2 \pi D_2^2}{4 L_2} \tag{51}$$

where $d_1$, $D_1$ represents the inner and outer diameters of rod 1; $D_2$ represents the diameter of rod 2; $L_1$, $L_2$ represents the lengths of rod 1 and rod 2.

The drive branch stiffness matrix is established as:

$$K_q = \left( K_s^{-1} + K_m^{-1} + K_{L1}^{-1} + K_{L2}^{-1} \right)^{-1} \tag{52}$$

The axial tensile stiffness of four driving branches can be calculated as:

$$\begin{cases} K_{CD} = 54,138 \\ K_{CE} = 36,867 \\ K_{AD} = 21,528 \\ K_{BF} = 54,138 \end{cases} \tag{53}$$

The force analysis of the central connecting rod is carried out. The central branch is constrained by two forces acting on the center of the ball pair. The constraint force $F_{CF1}$ is along the direction of the central branch, and the constraint force $F_{CF2}$ is parallel to the direction of the rotating pair. The angle between constraint $F_{CF1}$ and $F_{CF2}$ is denoted as $\theta_{CF}$. The constraint forces of the central branch are shown in Figure 13.

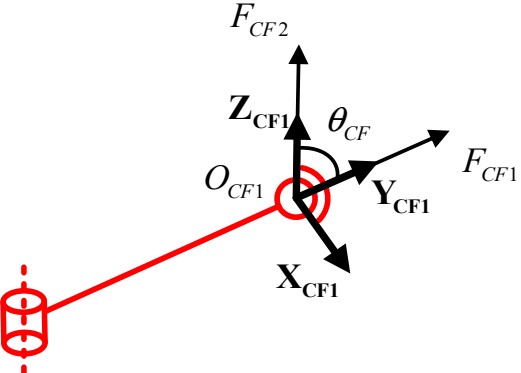

**Figure 13.** Analysis of constraint forces of the central branch.

The constraint force $F_{CF1}$ generates tensile deformation on the central rod, so its axial tensile stiffness is obtained as:

$$K_{CF1} = \frac{E_{CF} \pi d_{CF}^2}{4 L_{CF}} \tag{54}$$

The constraint force $\boldsymbol{F}_{CF2}$ generates tensile and bending deformation on the central rod. Thus, $F_{CF2}sin(\theta_{CF})$ generates bending deformation, and $F_{CF2}cos(\theta_{CF})$ generates tensile and compression deformation. Therefore, the axial stiffness and bending stiffness of the central connecting rod can be calculated as:

$$K_{CF2} = \frac{E_{CF}\pi d_{CF}^2}{4L_{CF}\cos(\theta_{CF})} \tag{55}$$

$$K_{CF3} = \frac{3E_{CF}I_{CF}}{L_{CF}^3\sin(\theta_{CF})} \tag{56}$$

$$I_{CF} = \frac{\pi d_{CF}^4}{64} \tag{57}$$

where $K_{CF2}$ represents the tensile stiffness of the central connecting rod, $K_{CF3}$ represents the bending stiffness of the central connecting rod, and $I_{CF}$ represents the moment of inertia of the central connecting rod.

By applying the structural parameters, the stiffness of the central branch is obtained as:

$$\begin{cases} K_{CF1} = 107,861 \\ K_{CF2} = 115,535 \\ K_{CF3} = 19,689 \end{cases} \tag{58}$$

Therefore, the branch stiffness matrix of the wing morphing mechanism can be calculated as:

$$\boldsymbol{K_l} = \begin{bmatrix} K_{CD} & 0 & 0 & 0 & 0 & 0 \\ 0 & K_{CE} & 0 & 0 & 0 & 0 \\ 0 & 0 & K_{AD} & 0 & 0 & 0 \\ 0 & 0 & 0 & K_{BF} & 0 & 0 \\ 0 & 0 & 0 & 0 & K_{CF1} & K_{CF3} \\ 0 & 0 & 0 & 0 & 0 & K_{CF2} \end{bmatrix} \tag{59}$$

The stiffness matrix of the wing morphing mechanism is obtained by the Jacobian matrix as:

$$\boldsymbol{K} = \boldsymbol{J}^{\mathrm{T}}\boldsymbol{K_l}\boldsymbol{J} \tag{60}$$

Then, based on the stiffness theoretical model of the wing morphing mechanism, the mapping relationship between the end position deformation and the external force is obtained. The structural parameters are substituted into the theoretical stiffness model to obtain the theoretical deformation. The force along the *z*-axis of 100 N is applied to the origin position of the moving coordinate system of the mechanism. ANSYS is employed to simulate the wing morphing mechanism to obtain the simulation deformation at the end position of the mechanism.

When the external load is the force of 100 N along the *z*-axis and the material is structural steel, the theoretical deformation is obtained as:

$$\boldsymbol{D}_{LP}^{ST} = \begin{bmatrix} -0.0051 & -0.0068 & 0.0102 \end{bmatrix} \tag{61}$$

When the external load is the force of 100 N along the *z*-axis and the material is 7075 aviation aluminum alloy, the theoretical deformation is obtained as:

$$\boldsymbol{D}_{LP}^{AL} = \begin{bmatrix} -0.0298 & -0.0372 & 0.0305 \end{bmatrix} \tag{62}$$

The model was simulated and verified with ANSYS without considering the deformation of the tensegrity structure, so it was set as a rigid body. The rigid rod in the tensegrity structure and the connecting rod in the driving branch of 4SPS-RS parallel mechanism are both two-force rods. Therefore, it was set as the rod unit. Each part is meshed based on tetrahedral, hexahedral and other basic units. By comparing the deformation of 1 mm,

3 mm, 5 mm mesh size, when the mesh size is 1 mm, the deformation will not change greatly. A variety of simulation results were obtained. By comparing the deformation, the three closest meshing methods were obtained. The difference of deformation was within 0.01 mm. The grid can be considered convergent. The NODE point was established at the centroid position of the moving coordinate system, and the deformation was measured to obtain the simulation results. As shown in Figure 14.

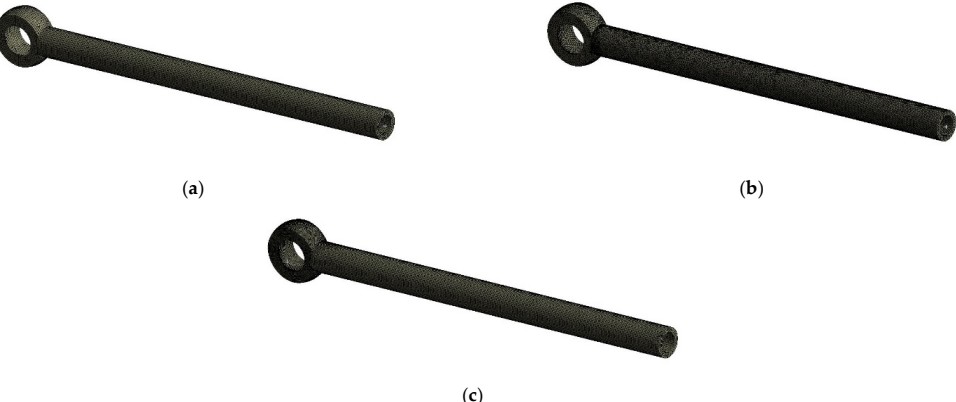

(**a**)

(**b**)

(**c**)

**Figure 14.** Three grid generation methods for parts: (**a**) hexahedral mesh generation method; (**b**) tetrahedron mesh generation methods with different sizes; (**c**) tetrahedron and hexahedron mesh generation method.

When the external load is the force of 100 N along the *z*-axis and the material is structural steel, the simulation deformation is shown in Figure 15.

$$D_{FP}^{ST} = \begin{bmatrix} -0.0045 & -0.0079 & 0.0108 \end{bmatrix} \tag{63}$$

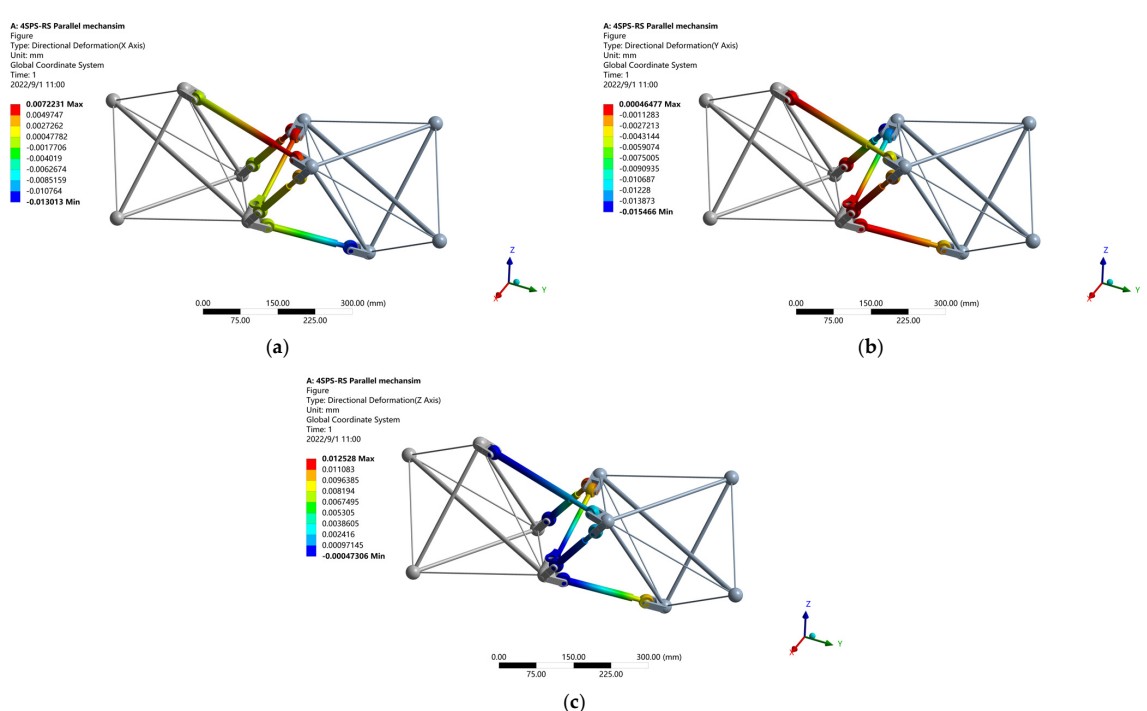

(**a**)

(**b**)

(**c**)

**Figure 15.** When the external load is the force of 100 N along the *z*-axis and the material is structural steel, simulation deformation of 4SPS-RS parallel mechanism in different coordinate directions: (**a**) *x*-axis; (**b**) *y*-axis; (**c**) *z*-axis.

When the external load is the force of 100 N along the *z*-axis and the material is 7075 aviation aluminum alloy, the simulation deformation is shown in Figure 16.

$$D_{FP}^{AL} = \begin{bmatrix} -0.0271 & -0.0366 & 0.0294 \end{bmatrix} \qquad (64)$$

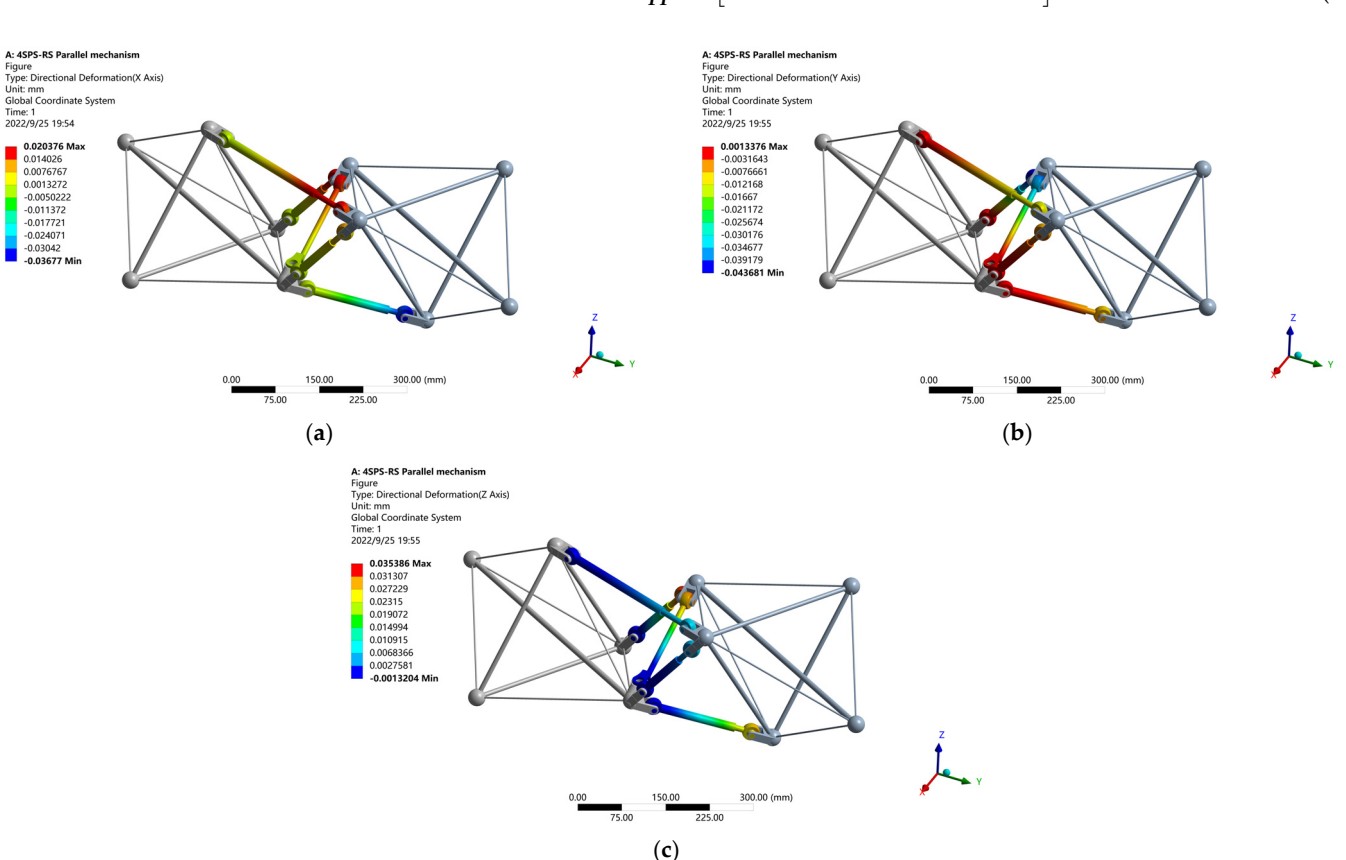

(**a**)

(**b**)

(**c**)

**Figure 16.** When the external load is the force of 100 N along the *z*-axis and the material is 7075 aviation aluminum alloy, simulation deformation of 4SPS-RS parallel mechanism in different coordinate directions: (**a**) *x*-axis; (**b**) *y*-axis; (**c**) *z*-axis.

The error between the theoretical value and the simulation value was within 8%, which verifies the correctness of the stiffness theoretical model.

According to the mission requirements, the variation range of sweep is $[-\pi/9 \sim 0]$, the variation range of twist is $[-\pi/9 \sim \pi/9]$, the variation range of spanwise bending is $[-\pi/9 \sim \pi/9]$, and the variation range of span is [0~50 mm]. The external force of 1000 N along the *z*-axis is applied at the origin position of the moving coordinate system. When the wing moves along the span direction to the minimum and maximum position, two parameters of the sweep, twist, and spanwise bending are changed. The total deformation along the coordinate axis of the wing morphing mechanism is obtained.

Then, any two parameters are changed at the minimum span, and the relationship between the deformation of the 4SPS-RS mechanism and the change of parameters is obtained. As shown in Figure 17.

Finally, any two parameters are changed at the maximum span, and the relationship between the deformation of the 4SPS-RS mechanism and the change of parameters is obtained. As shown in Figure 18.

By analyzing the deformation law in a large-lift environment, it was found that the wing morphing mechanism had a high degree of stiffness and bearing capacity, which could meet the morphing and bearing requirements during flight.

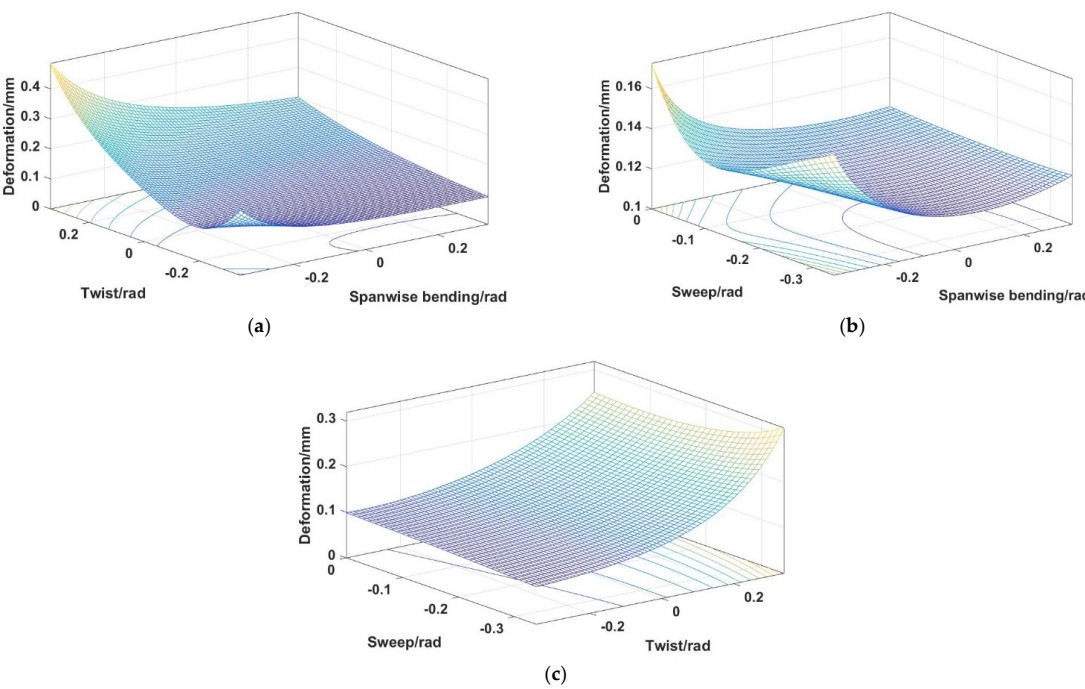

**Figure 17.** The relationship between the deformation of the 4SPS-RS parallel mechanism and sweep, twist, spanwise bending at the minimum span: (**a**) the relationship between the deformation and twist, spanwise bending; (**b**) the relationship between the deformation and sweep, spanwise bending; (**c**) the relationship between the deformation and sweep, twist.

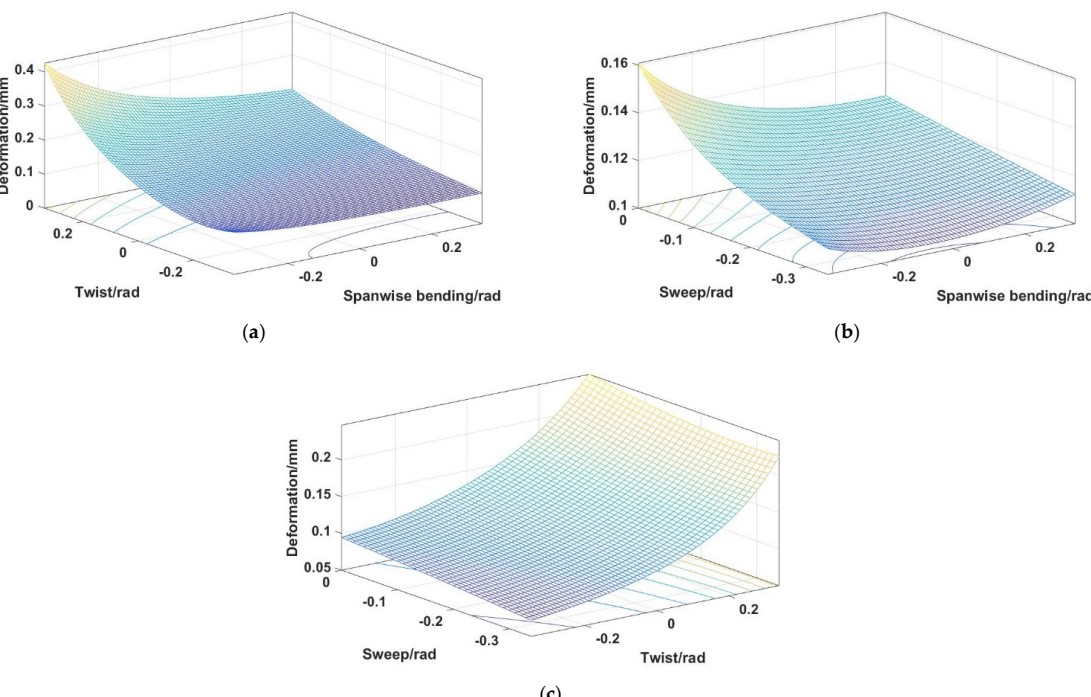

**Figure 18.** The relationship between the deformation of the 4SPS-RS parallel mechanism and sweep, twist, spanwise bending at the maximum span: (**a**) the relationship between the deformation and twist, spanwise bending; (**b**) the relationship between the deformation and sweep, spanwise bending; (**c**) the relationship between the deformation and sweep, twist.

## 5. Discussion and Conclusions

The design goal of a morphing wing is the pursuit of higher bearing capacity and lighter weight. The wing morphing mechanisms based on parallel mechanism used an 8SPS parallel mechanism to complete a six degrees of freedom wing morphing movement. It could achieve wing morphing with more degrees of freedom, but it increases the mass of the wing due to its redundant driving mode. The 4SPS-RS is a non-overconstrained parallel mechanism. It satisfies morphing motions and also has fewer drives. The existing wing span morphing technology used a ball screw mechanism, a crank slider mechanism and a plane folding link mechanism. Because these drive mechanisms cannot bear the external load, it will increase the quality of the wing. The wing morphing mechanism proposed in this paper adopts a distributed bearing integrated structure. This can not only complete wing morphing movements, but also withstand external loads. The existing wing twist morphing used modular structure assembly technology. Its assembly is simple and the weight of the structure is light. Due to structural limitations, it can only achieve one wing morphing motion. The morphing wing basic unit proposed in this paper consists of a trigonal bipyramidal tensegrity structure and a 4SPS-RS parallel mechanism. It can also be modularly assembled. The 4SPS-RS parallel mechanism can realize morphing motions of span, twist, sweep, and spanwise bending. Compared with the existing morphing wing structure, the morphing wing proposed in this paper can solve the shortcomings of the existing mechanism. Wings have the lowest weight loss and highest aerodynamic performance.

In this paper, a morphing wing composed of a trigonal bipyramidal tensegrity structure and a non-overconstrained parallel mechanism was proposed. The tensegrity structure consists of four rigid rods and six flexible cables. The optimal basic unit structure of the tension structure and the combinatorial unit layout were obtained by the shape-finding method. Then, a 4-DOF unconstrained parallel mechanism was designed based on the configuration synthesis method, which includes four SPS drive branches and one RS center constraint branch. The degree of freedom and inverse solution of the 4SPS-RS parallel mechanism was obtained based on the screw theory, and the Jacobian matrix of the parallel mechanism was established. Then the size optimization design of 4SPS-RS parallel mechanism was carried out. Afterward, the stiffness model of the tensegrity structure and the 4SPS-RS parallel mechanism was established. The influence of structural steel and 7075 aviation aluminum alloy on structural stiffness was analyzed. Finally, the correctness of the theoretical stiffness model was verified through ANSYS simulation. Furthermore, the relationship between the deformation of the 4SPS-RS parallel mechanism and sweep angle, torsion angle, spanwise bending, and span was obtained. In the static state, the morphing wing inherits the advantages of high stiffness and large bearing capacity of the tensegrity structure and the parallel mechanism. Meanwhile, it is lighter in weight due to the presence of flexible cables. In the process of morphing, it can complete the morphing motion of the wing with four degrees of freedom in changing the sweep, twist, spanwise bending, and span of the wing. Through the modular assembly and distributed drive, the morphing wing can perform smooth and continuous morphing locally and globally and avoid excessive local driving force.

**Author Contributions:** Conceptualization, Y.X. and J.S.; methodology, J.S.; software, X.L.; validation, J.S., X.L. and T.P.; formal analysis, J.S.; investigation, X.L.; resources, Y.X.; data curation, J.S.; writing—original draft preparation, J.S. and X.L.; writing—review and editing, Y.X.; visualization, J.S.; supervision, Y.X. and J.Y.; project administration, Y.X., J.Y. and Y.Z.; funding acquisition, Y.X. and Y.Z. All authors have read and agreed to the published version of the manuscript.

**Funding:** This research was funded by Projects supported by the National Natural Science Foundation of China, grant number 51875495 and U2037202. This research was funded by Projects supported by the Hebei Science and Technology Project, grant number 206Z1805G.

**Institutional Review Board Statement:** Not applicable.

**Informed Consent Statement:** Not applicable.

**Data Availability Statement:** Not applicable.

**Conflicts of Interest:** The authors declare no conflict of interest.

## Abbreviations

| | |
|---|---|
| $N$ | Node coordinate matrix |
| $n_i$ | Node coordinate |
| $C_B^T$ | Rigid rod connection matrix |
| $C_S^T$ | Flexible cable connection matrix |
| $B$ | Rigid rod position direction matrix |
| $S$ | Flexible cable position direction matrix |
| $A$ | Node force balance matrix |
| $s$ | Self-stress modal number |
| $m$ | Displacement modal number |
| $M$ | Degree of freedom |
| $h$ | The number of active components |
| $f_i$ | Degree of freedom of the $i$-th kinematic pair |
| $v$ | The number of overconstraints |
| $P_G$ | The coordinates of point $P$ in the fixed coordinate system |
| $P_H$ | The coordinates of point $P$ in the moving coordinate system |
| $T$ | Attitude transformation matrix |
| $Rot$ | Rotation transformation matrix |
| $Trans(\text{O}_G,\text{F})$ | Displacement transformation matrix from OG to F |
| $\$_{OH}$ | The joint motion spiral |
| $\theta_{i,j}$ | The amplitude of the $j$-th joint velocity of the $i$-th branch |
| $\$_{i,j}$ | The motion screw of the $j$-th joint of the $i$-th branch |
| $\$_{CF}^{f1}$ | Constraint force provided by CF branch |
| $Jy$ | Constraint Jacobian matrix |
| $Jq$ | Driving force Jacobian matrix |
| $J$ | 4SPS-RS mechanism Jacobian matrix |
| $Q_V$ | Drive branch velocity matrix |
| $Q_F$ | Driving force matrix |
| $f_{n3}^{b2}$ | The force at the nodes n3 caused by rod b2 |
| $K_S^{b2}$ | Tangent stiffness matrix |
| $D_{LT}^{ST}$ | Theoretical deformation of tensegrity structure when the material is structural steel |
| $D_{FT}^{ST}$ | Simulation deformation of tensegrity structure when the material is structural steel |
| $K_g$ | The axial stiffness of the screw |
| $K_m$ | The axial stiffness of the ball nut |
| $K_q$ | The drive branch stiffness |
| $K_{CF1}$ | The axial tensile stiffness of central rod |
| $K_l$ | The branch stiffness matrix |
| $K$ | Stiffness matrix |
| $D_{LP}^{ST}$ | Theoretical deformation of 4SPS-RS parallel mechanism when the material is structural steel |
| $D_{FP}^{ST}$ | Simulation deformation of 4SPS-RS parallel mechanism when the material is structural steel |

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
