# Peer review of "Morphing Wing Based on Trigonal Bipyramidal Tensegrity Structure and Parallel Mechanism"

_machines, doi:10.3390/machines10100930_

Round 1

Reviewer 1 Report

Comments:

1.   Please remove the (*) in equations 15 and 16.

2.  Why was a 4-DOF wing morphing mechanism chosen? The authors should justify their choice.

3. It is preferable to study the impact of the material choices on the structure's stiffness.

4. It is strongly recommended to include a section on the parallel mechanism's dimensional design.

5.  The results seem to be good but are not compared with other existing robots. Authors shall elucidate in detail the attractive results of the proposed technique, especially compared with the published literature (then, the conclusion of the article has to be revised).

6.  The relative works in the following references can be mentioned in the Introduction:

a)       J.-P. Merlet, Parallel Robots, 2nd edition Springer, 2006.

b)      Brahmia, A., Kelaiaia, R., Chemori, A., and Company, O. "On Robust Mechanical Design of a PAR2 Delta-Like Parallel Kinematic Manipulator." ASME. J. Mechanisms Robotics. February 2022; 14(1): 011001. https://doi.org/10.1115/1.4051360

7. There are many symbols. A list of symbols and abbreviations (nomenclature) should be given.

Author Response

Thank you for your valuable comments on the article, here is my answer.

  1. I have removed the (*) in equations 15 and 16.
  2. From the morphing direction, the morphing wing can be classified into two types: morphing along the wing span direction (span, twist, dihedral, spanwise bending and sweep); morphing along the flight direction of the wing (chord, camber and thickness). Morphing wings with multiple morphing motions can meet different mission requirements. This paper focuses on the morphing along the wing spanwise direction. According to the characteristics of morphing motions, changing the span of the wing is a translational motion along the x-axis direction; changing the twist of the wing is a rotational motion around the x-axis; changing the dihedral angle and spanwise bending of the wing is a rotational motion around the y-axis; changing the sweep of the wing is a rotational motion around the z-axis. Therefore, morphing motions along the spanwise direction can be summarized as translational motion along x-axis and rotational motion around x, y, z-axis. Compared with the traditional driving method, the distributed driving structure can achieve smooth and continuous morphing in the whole and local wings. In summary, it is proposed to design a wing morphing mechanism that can achieve many wing morphing motions and meet many mission requirements.
  3. The main materials of aircraft wing bearing structure are 7075 aviation aluminum alloy and structural steel. Therefore, the influence of two materials on stiffness is studied. Based on the stiffness theoretical model, the theoretical deformations of two materials are calculated. Based on ANSYS, the deformation of two materials is simulated. When the external load is the same, the deformation of 7075 aviation aluminum alloy is larger than that of structural steel. However, the weight of 7075 aviation aluminum alloy is lighter than that of structural steel. The combination of two materials is the best option, with high-strength structural steel selected at key connection points and 7075 aviation aluminum alloy used in the wing body.
  4. The dimension optimization design of 4SPS-RS parallel mechanism is carried out. According to the morphing mission requirements, the variation range of sweep is [-π/9~0], the variation range of twist is [-π/9~π/9], the variation range of spanwise bending is [-π/9~π/9], and the variation range of span is [0~50mm]. In order to make the 4SPS-RS parallel mechanism have better force performance. Driving force stability and balance driving force maximum value are taken as optimization objectives. The balanced driving force of the mechanism can be calculated based on the force Jacobian matrix.The three fixed nodes of the 4SPS-RS parallel mechanism form an equilateral triangle whose length is the side length of the tensegrity structure. Since the basic elements of tensegrity structures are identical, three active nodes form an identical equilateral triangle. The length of the equilateral triangle is Lt. The vertical height between the fixed surface and the active surface is Lp In the workspace, two sizes are optimized. The variation range of sizes is given. The variation range of Lt is [150~350]. The variation range of Lp is [100~300]. These sizes are discretized. The moving platform moved 50mm along y-axis, rotated 20 degrees around the x-axis and rotated 10 degrees around the y-axis and z-axis. The external loads are 100N force along the x-axis and y-axis. The relationship between the maximum equilibrium driving force and the size is obtained. Finally, the optimal size combination is obtained.
  5. This paper compares and analyzes wing morphing mechanisms in recent 3 years ' related literature. According to the advantages and disadvantages of existing mechanisms, the advantages and disadvantages of the morphing wing proposed in this paper are compared. The design goal of morphing wing pursues higher bearing capacity and lighter weight. Wing morphing mechanisms based on parallel mechanism used 8SPS parallel mechanism to complete six degree of freedom wing morphing movement. It can achieve wing morphing with more degrees of freedom, but it increases the mass of the wing due to its redundant driving mode. The 4SPS-RS is a non-overconstrained parallel mechanism. It satisfies morphing motions and also has less number of drives. The existing wing span morphing technology used ball screw mechanism, crank slider mechanism and plane folding link mechanism. Because these drive mechanisms cannot bear the external load, it will increase the quality of wing. Wing morphing mechanism proposed in this paper adopts distributed bearing integrated structure. Can not only complete wing morphing movements, but also withstand external loads. The existing wing twist morphing used modular structure assembly technology. Its assembly is simple and the weight of the structure is light. Due to structural limitations, it can only achieve one wing morphing motion. The morphing wing basic unit proposed in this paper consists of a trigonal bipyramidal tensegrity structure and a 4SPS-RS parallel mechanism. It can also be modularly assembled. The 4SPS-RS parallel mechanism can realize morphing motions of span, twist, sweep, and spanwise bending. Compared with the existing morphing wing structure, the morphing wing proposed in this paper can solve the shortcomings of the existing mechanism. Wings have the lowest weight loss and highest aerodynamic performance.
  6. I have added listed references to the introduction
  7. I have listed the symbol table in the appendix

Reviewer 2 Report

This paper proposes a morphing wing composed of a trigonal bipyramidal tensegrity structure and a non-overconstrained parallel mechanism. The tensegrity structure consists of four rigid rods and six flexible cables.

The authors described the optimal basic unit structure of the tension structure and the combinatorial unit layout. Then, a 4-DOF unconstrained parallel mechanism was proposed by the authors.

The degree of freedom and inverse solution of the 4SPS-RS parallel mechanism was obtained based on the screw theory, and the Jacobian matrix of the parallel mechanism was established. Afterwards, the stiffness model of the tensegrity structure and the parallel mechanism was established. An FE model was developed in ANSYS to verify the theoretical model and it was found to be within 8%.

The research presented in this paper is interesting and relevant. However, the reviewer would like to have more details of the FE model used to validate the theoretical model. Specific details about the element types and a mesh convergence study are required to establish the accuracy of the FE model. These elements are missing in the proposed paper.  

In sum, the paper contains elements of a solid work suitable for archival publication.  Hence, the reviewer recommends the acceptance after minor revision of the paper.

Author Response

Thank you for your valuable comments on the article, here is my answer.

The rigid rod in tensegrity structure and the connecting rod in driving branch of 4SPS-RS parallel mechanism are both two-force rod. Therefore, it is set as the rod unit. Each part is meshed based on tetrahedral, hexahedral and other basic units. By comparing the deformation of 1mm, 3mm, 5mm mesh size, when the mesh size is 1mm, the deformation will not change greatly. Get a variety of simulation results. By comparing the different meshing methods, the deformation error of three meshing methods is within 0.01 mm. The grid can be considered convergent. The NODE point is established at the centroid position of the moving coordinate system, and the deformation is measured to obtain the simulation results.

Round 2

Reviewer 1 Report

Figures 11, 12, 15, and 16 did not appear in the revised paper.